# Determining Effective Environmental Factors in the Distribution of Endangered Endemic Medicinal Plant Species Using the BMLR Model: The Example of Wild Celery (*Kelussia odoratissima* Mozaff., Apiaceae) in Zagros (Iran)

**DOI:** 10.3390/plants11212965

**Published:** 2022-11-03

**Authors:** Esfandiar Jahantab, Mohammad Reza Mahmoudi, Mohsen Sharafatmandrad, Vahid Karimian, Esmaeil Sheidai-Karkaj, Abdolvahab Khademi, Mohammad Reza Morshedloo, Christophe Hano, Jose M. Lorenzo

**Affiliations:** 1Department of Range and Watershed Management (Nature Engineering), Faculty of Agriculture, Fasa University, Fasa P.O. Box 74616, Iran; 2Department of Statistics, Faculty of Science, Fasa University, Fasa 74616, Iran; 3Department of Ecological Engineering, Faculty of Natural Resources, University of Jiroft, Jiroft P.O. Box 7867161167, Iran; 4Department of Nature Engineering, Faculty of Natural Resources, Yasouj University, Yasouj P.O. Box 75918, Iran; 5Department of Range and Watershed Management, Faculty of Natural Resources, Urmia University, Urmia P.O. Box 57561, Iran; 6Department of Mathematics and Statistics, University of Massachusetts, Amherst, Boston, MA 01003, USA; 7Department of Horticultural Science, Faculty of Agriculture, University of Maragheh, Maragheh P.O. Box 55187, Iran; 8Laboratoire de Biologie des Ligneux et des Grandes Cultures, INRA USC1328, Orleans University, CEDEX 2, 45067 Orléans, France; 9Centro Tecnológico de la Carne de Galicia, Rúa Galicia Nº 4, Parque Tecnológico de Galicia, San Cibrao das Viñas, 32900 Ourense, Spain; 10Área de Tecnoloxía dos Alimentos, Facultade de Ciencias, Universidade de Vigo, 32004 Ourense, Spain

**Keywords:** ecological factors, endangered species, *Kelussia odoratissima* Mozaff., mountainous rangeland

## Abstract

*Kelussia odoratissima* Mozaff. is a medicinal species native to Iran. The goal of this research was to determine the environmental factors important for the distribution of *K. doratissima* in Iran using BMLR modeling. Six random transects were established throughout the species’ habitat, and 220 quadrats with an area of 4 m^2^ were plotted. The canopy cover percentages of *K. doratissima* were estimated in each quadrat. Topographic factors, including elevation, slope, and aspect maps, were generated by creating DEM images. Land use, land evaluation, evaporation, temperature, and precipitation maps of the area were created accordingly. The data collected from the experiments were analyzed using the Minitab and R statistical packages. To determine the effect of the studied factors in the distribution of *K. doratissima*, we ran a set of backward multiple linear regressions. The results showed that the effects of evaporation, elevation, and slope were significant in the species’ distribution, with elevation having a positive effect and evaporation and slope showing negative effects. Further, elevation had the highest effect on distribution (greatest absolute value of beta at 9.660). The next most significant factors in the plant’s distribution were evaporation (beta = 8.282) and slope (beta = 0.807), respectively.

## 1. Introduction

Iran hosts a rich habitat for many medicinal and aromatic herbs, which have many benefits. Among the various plant families, Apiaceae is one that has more medicinal herbs and plants that can be used for industry [1]. Among the species in the Apiaceae family, wild celery (*Kelussia odoratissima* Mozaff.) can be named as a plant with non-fodder uses, such as medicinal, food, industrial, etc. applications, in addition to its use as fodder. This plant is a rare and endangered species that is indigenous to Iran and limited to some parts of the central Zagros Mountains in the southwest of Iran (including the Kohgiluyeh va Boyer Ahmad Province). This species is very beneficial to the local economy because the locals gather it and sell it in local and regional markets; unfortunately, its existence is threatened by utilization. In terms of life form, the wild celery species is a perennial plant in the Apiaceae family with a height reaching from 120 cm to 200 cm [2].

This species has a variety of uses. Pharmacologically, it has analgesic, sedative, anti-cough, anti-cancer, and anti-venom properties, and its flavonoid compounds are mainly accumulated in the seeds, stems, and inflorescences of the plant [3]. The green organs of the plant are harvestable in May. They have significant economic value and constitute a source of income for the local people. The dried organs can also be packaged and sold on the market.

In order to recognize the susceptible and hotspot areas for this plant’s growth and prevent its destruction, it is necessary to study the environmental factors controlling this plant’s spatial distribution. Recently, niche ecology has become one of the most important issues in plant protection studies [4]. Investigating the relationships between plant species and environmental factors has always been one of the main issues of study in plant ecology. The introduction of prediction mapping of plant species based on ecological nest models occurred in parallel with the development of related statistical methods and geographic information systems. Using these methods, the spatial distribution of plant species, based on the spatial distribution of environmental variables correlated with vegetation, is presented as a model in various forms of maps, tables, and diagrams. These models can play a prominent role in monitoring, evaluating, restoring, conserving, and sustainably developing rangeland ecosystems [5]. In addition, they are considered potential tools for learning about the causes of the distribution of species and the suitability of the habitat for plant species [6].

The presence of any plant species is influenced by environmental factors and inter-species relationships. It is possible that different environmental factors differently contribute to the presence of a particular plant species; therefore, if the environmental factors that control plant species’ distribution and the behavior of the species are determined, specific distribution models can be obtained [7]. Statistical models are used to express the relationship between the presence of the species and the relevant environmental factors. In these models, the response or dependent variables are the presence and absence of the target species, the predictor or independent variables are the environmental parameters, and the relationships between the variables are presented as statistical functions. Different statistical methods, such as regression and ordination, are used to define the relationships between the response and the predictor variables.

Using ordination methods, the relationships between all plant species and environmental factors can be analyzed and modeled simultaneously, while in a regression analysis, the relationship between each plant is examined separately and presented as a model. In vegetation ecology, regression models are used to estimate the optimum and tolerance ranges of plant species and to predict the reactions of species (e.g., abundance, presence, and absence) to environmental factors [8].

There are two types of regression models: parametric and nonparametric. In parametric regression, the relationship (linear, sigmoid, or Gaussian) between the response variable and the environmental factors is usually predetermined, while in nonparametric regression, there are no assumptions with respect to the shape of the response variable’s distribution.

In a study, Carter et al. [9] assessed the relationship between the presence and absence of *Aphelocoma coerulescens* and environmental factors using a logistic regression model. They evaluated their model using the kappa coefficient and the error matrix. Their results showed that the logistic regression model was highly accurate in predicting the occurrence of species. Lassueur and Randin [10] collected data on 117 plant species at 125 sites in southern Sweden. At each site, the elevation, slope, and aspect were measured. The probability of the occurrence of any plant was then predicted using logistic regression. The results showed that slope and aspect had the highest correlations with the occurrence of plant species. Some researchers investigated the relationships between 71 plant species and 11 environmental variables (altitude, light, and climatic variables) in the Alpine mountains using parametric regression. The results showed that the presence of all species is particularly dependent on environmental factors. Zare Chahuki et al. [11] used logistic regression modeling in Yazd rangelands in order to determine the relationship between the presence of plant species and environmental factors. The results of their study showed that soil properties, such as limestone, gravel, saturation moisture, gypsum, and electrical conductivity, were the most effective factors in determining vegetation types. Brown [12] studied the relationship between four plant species with topographic variables, slope, light, snow, and soil water potential using GLM (generalized linear models) and GAM (generalized additive models). He concluded that topographic variables were the most effective factors for the distribution of the studied species.

In light of the foregoing studies, it is of great importance to scientifically study *K. odoratissima*. To achieve this goal, the habitat of this species should first be identified so that further strategies can be developed for its conservation and reasonable use. In addition, to improve this species’ distribution and prevent its extinction, ecological factors affecting the distribution and the growth of this plant should be investigated.

## 2. Results

The first subsection presents descriptive statistics, including the means and standard deviations of the research variables. In Section 2, the statistical BMLR procedure is employed to investigate the effect of study factors on canopy cover. Table 1 shows the descriptive statistics of the measured research variables.

In this subsection, the importance of different environmental factors for the distribution of *K. odoratissima* (canopy cover of *K. odoratissima*) is investigated. In this research, canopy cover was the response variable. Further, the variables of rain, evaporation, elevation, slope, and aspect were continuous predictor variables. The general equation of MLR was presented as
$$Canopy\;{Cover}_{i}=\beta_{0}+\beta_{1}Rain_{i}+\ldots +\beta_{5}Aspect_{i}+\epsilon_{i}$$

The statistical BMLR model was applied using the R and Minitab statistical packages. First, all terms were entered and the fully entered model was run. Table 2 shows the results of the first BMLR run after removing $\beta_{0}$ (*p*-value > 0.05). As Table 2 indicates, when considering the effects of other variables, the effects of rain and aspect were not significant (*p*-value > 0.05).

Then, the most non-significant term (the term with the highest *p*-value) in the first run (rain) was removed, and the analysis was run again. Table 3 shows the results of the BMLR model after removing the rain term. As can be seen in Table 3, after controlling for the effects of other factors, the aspect predictor was not statistically significant.

Next, the most non-significant term in the second run (aspect) was removed, and the reduced model was run. Table 4 shows the results of the last BMLR run. As can be seen in Table 4, the effects of evaporation, elevation, and slope on the canopy cover of *K. odoratissima* were statistically significant. Elevation had a positive effect (positive coefficient), while evaporation and slope had negative effects (negative coefficient). Moreover, the factor of elevation had the greatest effect on the distribution of *K. odoratissima* (canopy cover) (the greatest absolute value of beta, 9.660). Following elevation, the factors of evaporation (beta = 8.282) and slope (beta = 0.807) had the next greatest effects, respectively (see Table 4).

The equation of the final and best BMLR model was
$$\hat{\mathrm{Canopy}\;\mathrm{Cover}}=-0.090\;\mathrm{Rain}+0.063\;\mathrm{Height}-0.611\;\mathrm{Slope}.$$

Next, the model’s goodness of fit was assessed by the coefficient of determination (R^2^), the adjusted coefficient of determination ($R_{adj}^{2}$), and the root mean square error (RMSE). A lower RMSE value and higher R^2^ and $R_{adj}^{2}$ values were regarded as showing a high goodness of fit level in the predictive model. As can be seen in Table 5, it seems that the BMLR modeled the canopy cover quite well based on the RMSE, R^2^, and $R_{adj}^{2}$ values.

The normality of the residuals was investigated using a probability plot and different normality tests (Anderson–Darling, Kolmogorov–Smirnov, and Shapiro–Wilk). As Figure 1 indicates, the points were close to the one-to-one line. Therefore, the normality assumption was satisfied. This assumption was also satisfied with different statistical tests (*p*-value > 0.05).

Figure 2 and Figure 3 illustrate the plots of residuals versus the observations’ order and the fitted values, respectively. As can be seen, the residuals were completely random around the horizontal axis; therefore, the independence and stability of the variances for the residuals were satisfied. Therefore, the BMLR nicely modeled the canopy cover.

## 3. Discussion

There are many studies on the effects of environmental factors on the distribution of endemic plant species [13,14,15,16,17]. A species’ distribution depends on a variety of factors, such as chemical, physical, and biological variables [18]. Ecologists have been investigating environmental factors controlling plant distribution and diversity for centuries [19]. Ecosystem development over time and changes in vegetation diversification in different ecological habitats are not random phenomena, but rather, are related to matrices of controlling factors [20]. In order to improve the management of vegetation, the ecological relationships between environmental factors, such as topography, climate, soil, vegetation, and organisms, should be clearly recognized [21,22]. Environmental factors, such as temperature, soil nutrients, and moisture, can affect the spatial distributions of plants and animals [23]. Therefore, detecting the variations in plant species’ distributions for the purpose of providing a logical perception of the environmental requirements of the species needed for successful ecological restoration is important [24]. On the other hand, knowledge of the ecological distribution of key species is the primary characteristic of the conservation strategy for its ecosystem [25,26]. Therefore, this research aimed to determine the environmental factors important for the distribution of *K. odoratissima* in Iran using BMLR modeling. Vegetation distribution models are more useful on a local scale, and based on our results, it is possible to determine potential habitats for plant species.

The results from our study showed that the effects of rain and aspect were not statistically significant. However, our results showed that the effects of evaporation, elevation, and slope on canopy cover were statistically significant. Elevation had a positive effect (positive coefficient), and evaporation and slope had negative effects (negative coefficients). Further, elevation had the greatest effect on canopy cover (with the greatest absolute value of beta, 9.660). Therefore, the largest factor of determination among the studied variables on the canopy cover of *K. odoratissima* was the elevation. Generally, *K. odoratissima* grows well in areas with a minimum elevation of 2500 m above sea level. With an increase or decrease in elevation, habitat conditions change, especially in terms of climate, and plants settle in an elevation range according to their ecological needs. Gradually, when we move from sea level to high areas, the type of soil and the related vegetation also changes. At low elevations, changes in plant communities are mostly related to soil characteristics, but in high-elevation areas, elevation plays a more important role in the changes in plant communities. The results obtained in our study are consistent with the results of Sakhavati et al. [27] that stated the elevation factor was the most important environmental factor in the establishment and distribution of the *Cerasus mahaleb* species. Following this factor, evaporation (beta= 8.282) and slope (beta = 0.807) had the next greatest effects, respectively. In this regard, previous studies reported that the most important factor affecting the distribution of *Pomatosace filicula* was predicted to be altitude [26]. Based on our results, *K. odoratissima* grows in heights of 2450 to 3000 m within the region of study. Researchers have reported that the optimum altitude for *Pomatosace filicula* was 4000–4500 m [28].

The evolution of the ecosystem and the dynamics of vegetation diversities in ecological habitats of the rangelands are not formed in a randomized manner, but rather, are formed as matrices of the most important environmental factors over time [20]. Plant species distribution over a high geographical range is controlled by climatic factors, mainly temperature and rainfall. In small ranges, the distribution of a species is related to edaphic factors [22]. Therefore, these environmental factors are important not only for detecting plant species distribution variations on a spatial scale but also for providing insight into the environmental requirements of the species that are needed for successful ecological restoration and the establishment of plantations [24]. Previous studies have shown that temperature and precipitation sustain the greatest impact on the distribution of *Daphne mucronata* species. Previous studies reported that soil variables, altitude, slope, aspect, and angle had a strong effect on species richness and plant distribution [29,30]. Generally, in mountainous areas, topographic characteristics, such as elevation, slope, and aspect, are the main factors influencing vegetation patterns. Researchers also considered the elevation factor to be more effective than the slope and aspect factors in changing the vegetation among the topographical factors [31]. As previous reports have stated, elevation has an important role in plant density [32,33].

Elevation was one of the controlling factors of *K. odoratissima* distribution. Topographic factors are one of the main environmental factors that are closely related to vegetation communities, a finding which was discovered in many previous studies [34]. The *K. odoratissima* habitat is located in the uneven and impassable Delafruz snow-capped mountains in central Zagros (Iran), and it seems that, in mountainous regions, topographical factors (especially elevation) play an important role in plant species distribution. It seems that on steep slopes with extensive rock outcrops, topographical factors have a greater effect on plant distribution than edaphic factors. In contrast, edaphic factors have a greater effect on gentle slopes and relatively thick soil over rock [35]. Schumann et al. [36] showed that elevation and precipitation are the most influential factors in the potential distributions of richness. In contrast, in arid and semi-arid ecosystems, studies show that soil properties play an important role in the distribution of plant species [37]. Previous studies reveal that some plant species can move to higher elevations or latitudes in response to a warming climate [28,38,39].

The evidence is clear that the *K. odoratissima* species has been severely damaged and is threatened by extinction. The main cause of *K. odoratissima* damage and loss can be attributed to its multi-purpose use as food, fodder, medicine, and for beekeeping. In other words, the plant has a variety of both human and animal utilizations that have resulted in excessive exploitation. According to the IUCN classification, species fall into four categories: endangered, vulnerable, lower risk, and data deficient [40]. According to available evidence, mountain celery is one of the endangered species (En). Endangered species will be in danger of extinction in the near future, worldwide, with a very high level of risk. If more serious protective actions and enclosures are not implemented, the species will probably fall out of the realm of nature and fall among the extinct species. Hence, considering the importance of the species’ susceptibility to extinction, it is necessary to protect its habitat. In addition, due to its importance and uses (in foods, forage, and medicine), it is necessary to develop this plant in suitable and proper areas. Regarding endangered species, it has been reported in previous studies that human activities can lead to the extinction of plant species [41,42]. Based on our results, *K. odoratissima* is being severely destroyed, and thus, is endangered. The main reasons for the destruction of this species are the multi-objective uses of this plant, such as its consumption as food, herbage, medicinal applications, bee breeding, and so on. In this regard, [43] reported that the vegetation of the Suruç (Şanlıurfa, Turkey) is being destroyed due to agricultural activities and herding. The *K. odoratissima* plant has many uses in the food and pharmaceutical industries. Many studies have been conducted on the traditional uses of this plant species [44,45,46,47,48]. In general, due to the major uses (in foods, forage, and medicine) of the *K. odoratissima* plant and its likely extinction, the protection of its habitat is necessary and the development of the habitat of the species in suitable areas is urged.

## 4. Materials and Methods

### 4.1. Study Area

The study area was the rough, snowy Delafrooz Mountain, which is located in the northwest of the Kohgiluyeh va Boyer-Ahmad Province, bordered by the Chaharmahal va Bakhtiari and Khuzestan provinces (50°18′–50°23′ and 1°23′–31°27′).

The climate in this area is semi-humid with a mean annual precipitation rate of 865 mm. The wet period lasts from late October to mid-May, and the dry period lasts from late May to mid-September. The mean monthly minimum and maximum temperatures are 2.6 °C and 27.1 °C, respectively. The average temperature is 15 °C. Most precipitation (about 75%) falls as snow, and the rest falls in the form of rain. The soils in the study area are generally shallow and non-uniform, with relatively high stone outcrops. The soil texture was loam and sandy loam. The organic carbon concentration of the soil was 3.25%. This geological region includes Asmari, Pabedeh, and Gourpi formations. The study area was located at heights from 2100 to 3000 m. The vegetation and flora of the study area are part of the dry forests of the Irano-Turanian region. The dominant plant types are Prangos-Kelussia and Prangos-Ferulago. Some other plant species include *Artemisia aucheri* Boiss., *Astragalus sp*, *Carduus arabicus* Jacq., *Daphne mucronata* Royle, and *K. odoratissima*.

### 4.2. Data Collection

Two areas were considered for sampling: current and former habitats of *K. odoratissima*. The current habitat (the area where the plant grows now) and the former habitat of *K. odoratissima* (the area where the plant grew previously and does not exist anymore) were determined through local experts, public guidance, and field surveys.

Six transects with different lengths (500, 500, 350, 300, 300, and 250 m) were randomly laid in each sampling area (the current and former habitats of *K. odoratissima*). It is worth noting that the length of the transects was determined according to the geomorphology of the region (mountainous and rocky). In other words, the difference in the transects’ lengths was due to the difference in the region area and the form of the land. Next, 2 × 2 m quadrats were established with 10 m intervals along each transect. In each quadrat, all plant species were listed, and their densities and canopy coverages were recorded.

In the middle of each transect, soil depth was measured and a sample was taken (6 samples for each region). Soil samples were transferred to the laboratory to determine the organic carbon and soil texture. The soil’s organic carbon was determined using the oxidation method [49]. Further, soil texture was determined using the hydrometer method. The position of the quadrats and transects were recorded by GPS.

### 4.3. Data Analysis

The data gathered from the experiments were analyzed using the Minitab and R statistical packages. To determine the effect of factors on the distribution of *K. odoratissima*, the researchers ran a set of backward multiple linear regressions.

Multiple linear regression (MLR) is a flexible method of data analysis that may be appropriate whenever a quantitative variable (as the dependent or response variable) is to be examined in relation to any other factors $X_{1},X_{2},\ldots ,X_{k}$ (as the independent or explanatory or predictor variables). The general equation of MLR is represented by
$$Y_{i}=\beta_{0}+\beta_{1}X_{1i}+\beta_{2}X_{2i}+\ldots +\beta_{k}X_{ki}+\varepsilon_{i}$$
for n observations i=1,…,n, where $\beta_{0},$, …, $\beta_{k},$ are model parameters (coefficients) and $\varepsilon_{i},i=1,\ldots ,n,$ are the random components of the model, which follow independent normal distributions with a mean value of 0, and a variance of $\sigma^{2}$.

The dataset is used to estimate the coefficients $\beta_{0},$ $\beta_{1}$, …, $\beta_{k}$. The general equation of the predictive MLR model is presented as
$${\hat{Y}}_{i}=b_{0}+b_{1}X_{1i}+b_{2}X_{2i}+\ldots +b_{k}X_{ki}$$
where, $b_{0},$ $b_{1}$, …, $b_{k},$ are estimations of the model parameters, and ${\hat{Y}}_{i}$ is the predicted value of $Y_{i}$.

The MLR model can be rewritten in matrix form as
Y=Xβ+ε
where $Y={\left(y_{1},\ldots ,y_{n}\right)}^{T}$ is a vector of responses, X is a n×(k+1) full rank design matrix with the first column given as ${\left(1,\ldots ,1\right)}^{T}$ and the $l^{th}\left(2\leq l\leq k+1\right)$ column given as ${\left(x_{l-1,1},\ldots ,x_{l-1,n}\right)}^{T}$, $\beta ={\left(\beta_{0},\ldots ,\beta_{k}\right)}^{T}$ is a vector of unknown parameters, and $\varepsilon ={\left(\varepsilon_{1},\ldots ,\varepsilon_{n}\right)}^{T}$ is a vector of random errors. Moreover, $\hat{Y}=Xb$, where $\hat{Y}={\left({\hat{y}}_{1},\ldots ,{\hat{y}}_{n}\right)}^{T}$ is a vector of predicted values, and $b={\left(b_{0},\ldots ,b_{k}\right)}^{T}$ is a vector of coefficients.

We note that in the model without interception ($\beta_{0}=0$), the column ${\left(1,\ldots ,1\right)}^{T}$ should be removed from matrix X.

The ordinary least squares (maximum likelihood) estimation of the coefficient vector β is given as
$$b={\left(X^{T}X\right)}^{-1}X^{T}Y.$$

This is a common approach, and it assumes that there are enough measurements to say something meaningful about β.

As can be seen, the MLR model contains the linear effects of $X_{1},X_{2},\ldots ,X_{k}$. However, perhaps some of these effects are not significant (*p*-value > 0.05). In this case, the backward method (BMLR) is used and, step by step, the not-effective parameters are removed. The final model has parsimonious parameters and maximum accuracy.

## 5. Conclusions

*K. odoratissima* is a species native to Iran. This research was conducted to determine the environmental factors important for the distribution of *K. odoratissima* in Iran using BMLR modeling. Our results showed that the effects of the rain and aspect were not statistically significant. Based on the present research results, it was shown that the effects of evaporation, elevation, and slope on the plant’s distribution were statistically significant. Elevation had a positive significant effect, while evaporation and slope had negative significant effects. In addition, the elevation had the greatest effect on distribution (with the greatest absolute value of beta, 9.660). Following this factor, evaporation (8.282) and slope (0.807) had the next greatest effects, respectively. Generally, the species grows in elevations from 2500 m to 3000 m at the steepest slopes of the study region. The minimum temperature of the region is −15 °C and the maximum temperature is 25 °C. The region’s soil is generally shallow with pebbles and it is semi-deep at some points. Our study addressed some aspects of the relationships between environmental factors and the distribution of the *K. odoratissima* species in Zagros, Iran. It was anticipated that this finding could be used as a tool for predicting the probability of the existence and the absence of this plant species in similar ecosystems.

## Figures and Tables

**Figure 1 plants-11-02965-f001:**
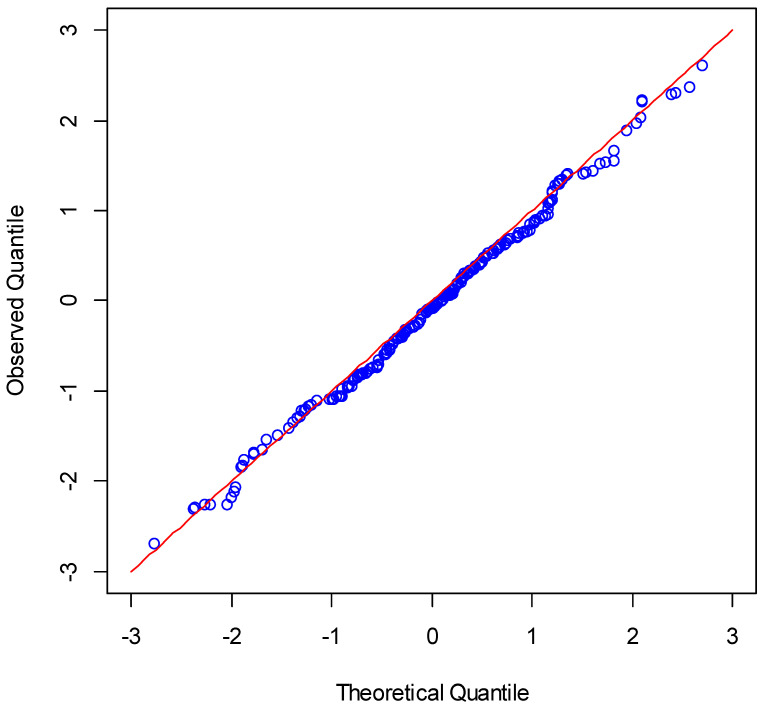
Normal probability plot of residuals for the BMLR model.

**Figure 2 plants-11-02965-f002:**
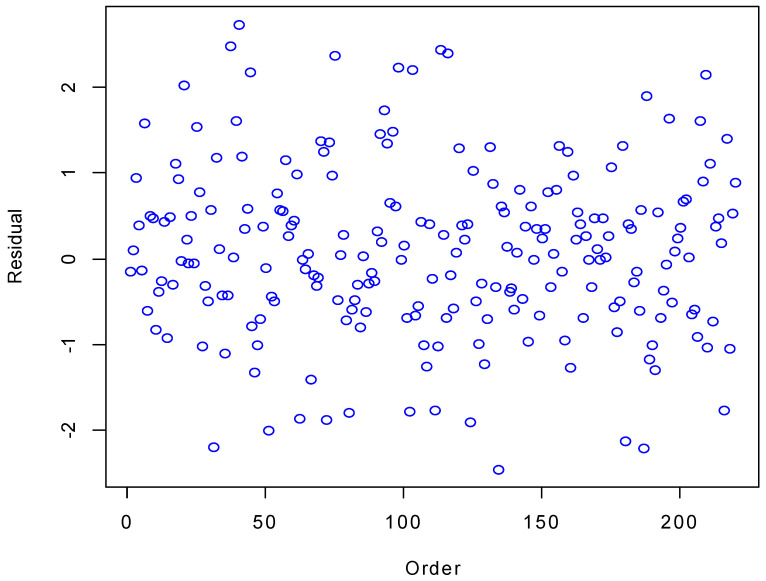
Plot of residuals versus the observation order for the BMLR model.

**Figure 3 plants-11-02965-f003:**
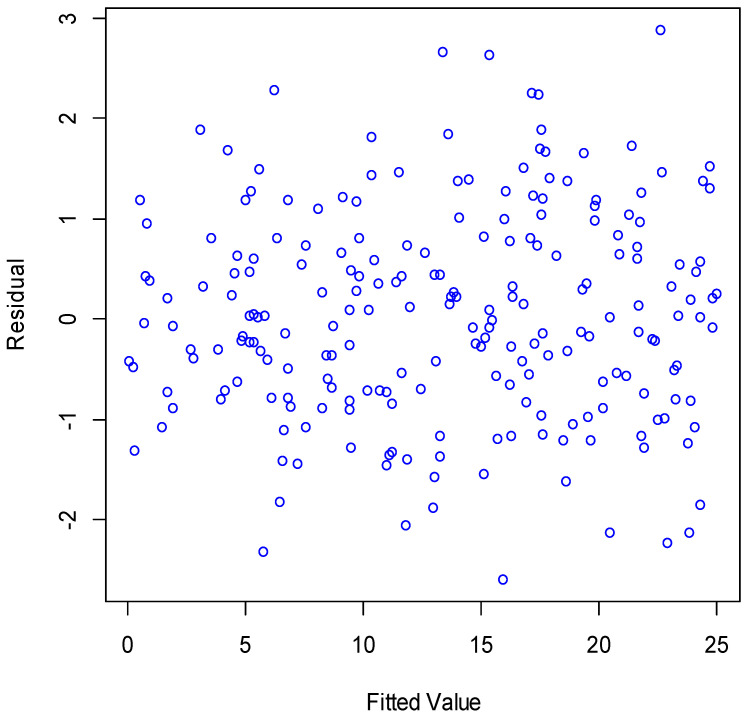
Plot of residuals versus the fitted values for the BMLR model.

**Table 1 plants-11-02965-t001:** Descriptive statistics of research variables.

Factor	N	Minimum	Maximum	Mean	Std. Deviation
Canopy Cover	220	0	85	10.90	14.240
Rain	220	826	828	827.17	0.773
Evaporation	220	1650	1678	1653.75	7.367
Elevation	220	2596	2853	2726.18	64.654
Slope	220	5	41	22.47	7.348
Aspect	220	31	225	160.60	41.009

**Table 2 plants-11-02965-t002:** First run of BMLR.

Model		Unstandardized Coefficients	Standardized Coefficients		t	*p*−Value
		B	Std. Error	Beta		
1	Rain	−0.039	0.339	0.169	−0.114	0.909
	Evaporation	−0.097	0.027	−8.939	−3.560	<0.001
	Elevation	0.069	0.018	10.430	3.832	<0.001
	Slope	−0.614	0.124	−0.810	−4.941	<0.001
	Aspect	−0.012	0.030	−0.113	−0.404	0.687

**Table 3 plants-11-02965-t003:** Second run of BMLR.

Model		Unstandardized Coefficients		Standardized Coefficients	t	*p*−Value
		B	Std. Error	Beta		
2	Evaporation	−0.097	0.027	−8.939	−3.560	<0.001
	Elevation	0.069	0.018	10.430	3.832	<0.001
	Slope	−0.614	0.124	−0.810	−4.941	<0.001
	Aspect	−0.012	0.030	−0.113	−0.404	0.687

**Table 4 plants-11-02965-t004:** Last run of BMLR.

Model		Unstandardized Coefficients		Standardized Coefficients	t	*p*−Value
		B	Std. Error	Beta		
3	Evaporation	−0.090	0.021	−8.282	−4.340	<0.001
	Elevation	0.063	0.013	9.660	4.986	<0.001
	Slope	−0.611	0.124	−0.807	−4.936	<0.001

**Table 5 plants-11-02965-t005:** Indexes for the goodness of fit for the fitted BMLR models.

R Square	Adjusted R Square	RMSE
0.469	0.461	13.147

## Data Availability

The data presented in this study are available on request from the corresponding author.

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
