# Peer review of "Determining Effective Environmental Factors in the Distribution of Endangered Endemic Medicinal Plant Species Using the BMLR Model: The Example of Wild Celery (Kelussia odoratissima Mozaff., Apiaceae) in Zagros (Iran)"

_plants, 2022, doi:10.3390/plants11212965_

Round 1

Reviewer 1 Report

Chapter numbering does not follow the journal style.
The keywords were ordered alphabetically.
Check if you have included enough details on statistics (number of replicates, statistical tests performed, presentation of average and standard deviation or error values, both in tables and graphs), and complete it if needed.
What are the original, novelty, or unique ideas behind this research as compared to previous research/other reported work? Why it is worth knowing?
The logic of the current introduction should be revised, objectives of this study need to be changed.
Graphic quality of many figures does not satisfy with publication standard. Provide the maps in high dpi resolution.
More articles should be discussed, especially among the international literature.

It is recommended that each referenced work is accompanied by a brief description of the key results and main conclusions. In this way, the inclusion of the cited work in the manuscript can be better justified.

Abstract is a bit long. Move the first several sentences in abstract to introduction would be better. The research gaps should be stated in introduction, not in abstract. You can tell the importance of the research question in abstract concisely.

The conclusions part should contain more useful and meaningful information, rather than just state what you have done. You should tell the reader what you have found.

Add more information about case study (rain, climate, soil etc).

Why did you use this method? What is the growth characteristic of the plant? Why didn't you use AHP? Please clearly indicate the novelty and scientific impacts in introduction section.

Author Response

Reviewer 1.

Chapter numbering does not follow the journal style.

RESPONSE: Section numbering changed to fit journal formatting.

The keywords were ordered alphabetically.

RESPONSE: correlations were added to the paper.

Check if you have included enough details on statistics (number of replicates, statistical tests performed, presentation of average and standard deviation or error values, both in tables and graphs), and complete it if needed.

RESPONSE: Thanks for your attention. This comment has been considered in revised version. The averages and the standard deviations of research variables are reported in Table 1. Moreover, the unstandardized coefficients and their standard error of BMLR runs are reported in Tables 2-4. Furthermore, as it can be seen in Section 4, the researchers run a set of backward multiple linear regression (BMLR) to investigate the importance of different environmental factors in the distribution of K. odoratissima (Canopy Cover of K. odoratissima). The number of replicates has been also added.

What are the original, novelty, or unique ideas behind this research as compared to previous research/other reported work? Why it is worth knowing?

RESPONSE: Thanks for your attention. This comment has been considered in revised version. Most of previous works considered coefficients of correlation and multiple linear regression (MLR) techniques to investigate the importance of different factors on the response variable. Coefficients of correlation consider the effect of just one factor and MLR considers the simultaneous effects of all significant and non-significant factors, while the BMLR technique considers the simultaneous effects of all factors and step by step excludes the non-significant factors.

Due to the fact that the K. odoratissima species is native to Iran and has many uses, it is important to know the environmental factors affecting the distribution of this species. Also, K. odoratissima species is at risk of extinction due to excessive exploitation, so knowing and introducing the factors affecting it can play an important role in its cultivation and domestication. Also, until now, study that determine the important environmental factors in the distribution of K. doratissima in Iran using BMLR modeling has not been conducted.

The logic of the current introduction should be revised, objectives of this study need to be changed.

RESPONSE: Done

Graphic quality of many figures does not satisfy with publication standard. Provide the maps in high dpi resolution.

RESPONSE: Thanks for your attention. This comment has been considered in revised version. The quality of the figures has been improved in revised version.

More articles should be discussed, especially among the international literature.

It is recommended that each referenced work is accompanied by a brief description of the key results and main conclusions. In this way, the inclusion of the cited work in the manuscript can be better justified.

RESPONSE: Done

Abstract is a bit long. Move the first several sentences in abstract to introduction would be better. The research gaps should be stated in introduction, not in abstract. You can tell the importance of the research question in abstract concisely.

RESPONSE: Done

The conclusions part should contain more useful and meaningful information, rather than just state what you have done. You should tell the reader what you have found.

RESPONSE: Done

Add more information about case study (rain, climate, soil etc).

RESPONSE: The average temperature is 15 Ëš C. Soil texture was loam and sandy loam. Organic carbon of soil was 3.25 %. This region geologically includes Asmari, Pabedeh and Gourpi formation. The study area located in the heights 2100 to 3000 m.

Why did you use this method? What is the growth characteristic of the plant? Why didn't you use AHP? Please clearly indicate the novelty and scientific impacts in introduction section.

RESPONSE: Thanks for your attention. As mentioned in previous comments, the BMLR technique considers the simultaneous effects of all factors and step by step excludes the non-significant factors. The BMLR method is a data-based approach while AHP is a survey-based approach. Data-based approaches are more powerful than survey-based approaches.

This comment has been considered in revised version. Most of previous works considered coefficients of correlation and multiple linear regression (MLR) techniques to investigate the importance of different factors on the response variable. Coefficients of correlation consider the effect of just one factor and MLR considers the simultaneous effects of all significant and non-significant factors, while the BMLR technique considers the simultaneous effects of all factors and step by step excludes the non-significant factors.

Reviewer 2 Report

I would like to present my positive opinion about the reviewed work.  The aim of this study was to Determining effective environmental factors in the distribution of endangered endemic medicinal plant species using the BMLR Model: the example of Wild celery (Kelussia odoratissima Mozaff., Apiaceae) in Zagros (Iran).

In the manuscript, the authors examine very important, and interesting issues related to the influence of environmental factors in the distribution of medicinal plants in the region of Iran. Currently, there is a great interest in natural medicine and herbal medicine. Harvesting herbs from unpolluted areas is therefore important.

The summary, work is very good. The results were broadly discussed. This work has very practical applications. I propose to accept this quality and publish it in the Plants.

Author Response

Reviewer 2.

General comments

I would like to present my positive opinion about the reviewed work.  The aim of this study was to Determining effective environmental factors in the distribution of endangered endemic medicinal plant species using the BMLR Model: the example of Wild celery (Kelussia odoratissima Mozaff., Apiaceae) in Zagros (Iran).

In the manuscript, the authors examine very important, and interesting issues related to the influence of environmental factors in the distribution of medicinal plants in the region of Iran. Currently, there is a great interest in natural medicine and herbal medicine. Harvesting herbs from unpolluted areas is therefore important.

The summary, work is very good. The results were broadly discussed. This work has very practical applications. I propose to accept this quality and publish it in the Plants.

RESPONSE: Thank you for your positive comments.

Reviewer 3 Report

Dear editor; The attached articled was checked. The manuscript contain interesting information about  Determining Effective Environmental Factors in the Distribution of Endangered Endemic Medicinal Plant Species Using the BMLR Model: The Example of Wild Celery (Kelussia odoratissima Mozaff., Apiaceae) in Zagros (Iran)

I think that this article well suits to your journal.

It is generally a good work. The scientific and presentation level of manuscript is high.

This paper may contain new and interesting ethnobotanical data from the high-biodiversity spot in the Iran

-I have only a few suggestions about the text The plant names can write short after first writing. Please, real the paper and correct them all.

Methodology is intelligible

References were cross checked?

In mat met: 2.6ËšC and  27.1ËšC,  : (Space ËšC) Leave space between words.

In Study area: Write the name of the authority to the end of the plant’s name.

How does the use of the plant species compare with those reported from other neighbouring regions, or regions on the rest of the country or neighbouring countries? It would be better if the article is supported with more index articles. Here are my suggestions; EXP: Botanical studies have considerably increased in recent years;

Korkmaz, M., KarakuÅŸ, S., Selvi, S., ÇakılcıoÄŸlu U., 2016. Traditional knowledge on wild plants in Üzümlü (Erzincan-Turkey). Indian Journal of Traditional Knowledge. 15 (4), 538-545.

Polat, R, Güner, B, Yüce-Babacan, E, ÇakılcıoÄŸlu, U., 2017. Survey of wild food plants for human consumption in Bingöl (Turkey). Indian Journal of Traditional Knowledge, 16 (3), 378-384.
At least one reference for 2021 must be added.

Yalçın, S. , Akan, H. & ÇakılcıoÄŸlu, U. (2021). Medicinal plants sold at herbal markets in Suruç district (Åžanlıurfa-Turkey). International Journal of Nature and Life Sciences, 5 (1), 40-51. doi: 10.47947/ijnls.932374

Güler, O, Polat R, Karakose M, CakılcıoÄŸlu U, Akbulut, S., 2021. An ethnoveterinary study on plants used for the treatment of livestock diseases in the province of Giresun (Turkey). South African Journal of Botany, 142, 53-62. 

Author Response

Reviewer 3.

General comments

Dear editor; The attached articled was checked. The manuscript contain interesting information about Determining Effective Environmental Factors in the Distribution of Endangered Endemic Medicinal Plant Species Using the BMLR Model: The Example of Wild Celery (Kelussia odoratissima Mozaff., Apiaceae) in Zagros (Iran)

I think that this article well suits to your journal.

It is generally a good work. The scientific and presentation level of manuscript is high.

This paper may contain new and interesting ethnobotanical data from the high-biodiversity spot in the Iran

I have only a few suggestions about the text The plant names can write short after first writing. Please, real the paper and correct them all.

RESPONSE: done

Methodology is intelligible

RESPONSE: Thank you so much for your kind comment. We improved this section.

References were cross checked?

RESPONSE: done; Yes, all references were checked

In mat met: 2.6ËšC and  27.1ËšC,  : (Space ËšC) Leave space between words.

RESPONSE: done

In Study area: Write the name of the authority to the end of the plant’s name.

RESPONSE: done

How does the use of the plant species compare with those reported from other neighbouring regions, or regions on the rest of the country or neighbouring countries? It would be better if the article is supported with more index articles. Here are my suggestions; EXP: Botanical studies have considerably increased in recent years;

Korkmaz, M., KarakuÅŸ, S., Selvi, S., ÇakılcıoÄŸlu U., 2016. Traditional knowledge on wild plants in Üzümlü (Erzincan-Turkey). Indian Journal of Traditional Knowledge. 15 (4), 538-545.

Polat, R, Güner, B, Yüce-Babacan, E, ÇakılcıoÄŸlu, U., 2017. Survey of wild food plants for human consumption in Bingöl (Turkey). Indian Journal of Traditional Knowledge, 16 (3), 378-384.

At least one reference for 2021 must be added.

Yalçın, S. , Akan, H. & ÇakılcıoÄŸlu, U. (2021). Medicinal plants sold at herbal markets in Suruç district (Åžanlıurfa-Turkey). International Journal of Nature and Life Sciences, 5 (1), 40-51. doi: 10.47947/ijnls.932374

Güler, O, Polat R, Karakose M, CakılcıoÄŸlu U, Akbulut, S., 2021. An ethnoveterinary study on plants used for the treatment of livestock diseases in the province of Giresun (Turkey). South African Journal of Botany, 142, 53-62.

RESPONSE: done

Round 2

Reviewer 1 Report

Most articles should be discussed. Specially among the international literature not national.

Discussion isn't acceptable. 

The paper does not follow journal style. 

Author Response

Thank you so much for your comments.

Now, the manuscript has been improved according to your suggestions

Round 3

Reviewer 1 Report

-Discussion is so weak.

-Conclusions too short.

  1. Restate your hypothesis or research question.
  2. Restate your major findings.
  3. Tell the reader what contribution your study has made to the existing literature.
  4. Highlight any limitations of your study.
  5. State future directions for research/recommendations

Author Response

-Discussion is so weak.
This section was improved and more references was discussed.  Some of the sentences are as follow: Also, Elevation had the greatest effect on Canopy Cover (with the greatest absolute value of Beta, 9.660). Therefor, the most determining factor among the studied variables on the Canopy Cover of K. odoratissima is Elevation. Generally, K. odoratissima grows well in areas with a minimum Elevation of 2500 meters above the sea level. With the increase or decrease in Elevation, habitat conditions change, especially in terms of climate, and plants settle in an Elevation range according to their ecological needs. Gradually, when we move from sea level to high areas, the type of soil and related vegetation also changes, and at low elevation, changes in plant communities are mostly related to soil characteristics, but in high elevation areas, elevation plays a more important role in the changes of plants communities. The obtained results in our study are consistent with the results of Sakhavati et al. [27] that stated the elevation factor was the most important environmental factor in the establishment and distribution of Cerasus mahaleb species.

-Conclusions too short.

 This section was improved as well.

K. odoratissima is a species native to Iran. So, this research was conducted to determine the important environmental factors in the distribution of K. odoratissima in Iran using BMLR modeling. Our results showed that the effects of Rain and Aspect were not statistically significant. Based on the present research results, it was shown that the effects of Evaporation, Elevation and Slope on the plant distribution were statistically significant. Elevation had a positive significant effect while Evaporation and Slope had negative significant effects. In addition, the Elevation had the greatest effect on distribution (with the greatest absolute value of Beta, 9.660). Following this factor, Evaporation (8.282) and Slope (0.807) had the next greatest effects, respectively. Generally, the species grows in the Elevation 2500 m to 3000 m at the steepest slopes of the study region. Minimum temperature of the region is -15ËšC and the maximum temperature is 25ËšC. The region soil is generally shallow with pebbles and it is semi-deep in some points. Our study addressed some aspects of the relationships between environmental factors and the distribution of K. odoratissima species in Zagros,Iran. It was anticipated that this finding could be used as a tool for predicting the probability of existence and the absence of this plant species in similar ecosystems.”

  1. Restate your hypothesis or research question.
  2. Restate your major findings.
  3. Tell the reader what contribution your study has made to the existing literature.
  4. Highlight any limitations of your study.
  5. State future directions for research/recommendations

 The manuscript was revised accordingly.

Round 4

Reviewer 1 Report

Does the abstract summarize the paper's objectives, main thrust and major conclusions? Please consider whether or not the Abstract conveys clearly the purpose of the study, provides a balanced and accurate depiction of the key findings, and addresses the implications of the work for spatial information Science. Could a person read the abstract and get a clear sense of what the article will be about? Will the key words enable other professionals to locate the work with the search engines commonly used by academic libraries? What about the conclusion? Does the manuscript give a sense of revisiting the main ideas briefly? Does it give the reader a feeling that all of the ideas have been tied together?

Check if you have included enough details on statistics (number of replicates, statistical tests performed, presentation of average and standard deviation or error values, both in tables and graphs), and complete it if needed.

Check if you have included self-explanatory captions for all tables and graphs, and complete and/or correct it if needed.

Check your whole manuscript for any typo and for English expression, in order to prevent this kind of mistakes.

More articles should be discussed, especially among the international literature.

What are the original, novelty, or unique ideas behind this research as compared to previous research/other reported work? Why it is worth knowing?

-All references should be APA style based.

Chapter numbering does not follow the journal style.

Author Response

Thank you very much for reviewing our manuscript. We appreciated the reviewer’s comments in order to improve the manuscript. Therefore, we have revised the manuscript amended with the changes requested by the reviewers. Below, you will find in track changes applied according to reviewer’s comments.

Comments and Suggestions for Authors                              

Does the abstract summarize the paper's objectives, main thrust and major conclusions? Please consider whether or not the Abstract conveys clearly the purpose of the study, provides a balanced and accurate depiction of the key findings, and addresses the implications of the work for spatial information Science. Could a person read the abstract and get a clear sense of what the article will be about? Will the key words enable other professionals to locate the work with the search engines commonly used by academic libraries? What about the conclusion? Does the manuscript give a sense of revisiting the main ideas briefly? Does it give the reader a feeling that all of the ideas have been tied together?

 Our reply# thanks for your attention; we revised the abstract and conclusion accordingly; you can see in below.

Introduction: Kelussia odoratissima Mozaff. is a medicinal species native to Iran. The goal of this research was to determine the important environmental factors in the distribution of K. doratissima in Iran using BMLR modeling. Material and Methods: Six random transects were established throughout the species habitat and 220 quadrats with the area of 4m2 were plotted. Canopy cover percentages of K. doratissima were estimated in each quadrat. Topographic factors, including elevation, slope and aspect maps were generated by creating DEM image. Land use, land evaluation, evaporation, temperature and precipitation maps of the area were accordingly created. The data collected from experiments were analyzed using the Minitab and R statistical packages. To determine the effect of factors in the distribution of K. doratissima, we ran a set of backward multiple linear regressions. Results:The results showed that the effects of evaporation, elevation, and slope were significant in the species distribution with elevation having a positive effect and evaporation and slope showing negative effects. Also, elevation had the highest effect on distribution (greatest absolute value of Beta of 9.660). The next most significant factors in the plant distribution were Evaporation (Beta = 8.282) and Slope (Beta = 0.807), respectively. Conclusion: Generally, K. odoratissima grows well in areas with a minimum elevation of 2500 meters above the sea level. Our results showed that, elevation is the most determining factor among the studied variables on the Canopy Cover and distribution of K. odoratissima.  Finally, the results provide useful information’s for conservation strategies of the endangered species as well as domestication and cultivations of the plant in suitable climatic conditions. 

Check if you have included enough details on statistics (number of replicates, statistical tests performed, presentation of average and standard deviation or error values, both in tables and graphs), and complete it if needed.

 Our reply# thanks for your attention; done; as you can see our tables have included enough details on statistics (number of replicates, statistical tests performed, presentation of average and standard deviation or error values. this comment has been considered in revised version. The averages and the standard deviations of research variables are reported in Table 1. Moreover, the unstandardized coefficients and their standard error of BMLR runs are reported in Tables 2-4. Furthermore, as it can be seen in Section 4, the researchers run a set of backward multiple linear regression (BMLR) to investigate the importance of different environmental factors in the distribution of K. odoratissima (Canopy Cover of K. odoratissima). The number of replicates has been also added.

Check if you have included self-explanatory captions for all tables and graphs, and complete and/or correct it if needed.

 Our reply# thanks for your attention; done;

Table 2. The results of the first run of BMLR

Table 3. The results of the second run of BMLR

Table 4. The results of the last run of BMLR

Check your whole manuscript for any typo and for English expression, in order to prevent this kind of mistakes.

 Our reply# thanks for your attention; done;

More articles should be discussed, especially among the international literature.

  Our reply# thanks for your attention; done;

What are the original, novelty, or unique ideas behind this research as compared to previous research/other reported work? Why it is worth knowing?

 Our reply# thanks for your attention; done; this comment has been considered in revised version. Most of previous works considered coefficients of correlation and multiple linear regression (MLR) techniques to investigate the importance of different factors on the response variable. Coefficients of correlation consider the effect of just one factor and MLR considers the simultaneous effects of all significant and non-significant factors, while the BMLR technique considers the simultaneous effects of all factors and step by step excludes the non-significant factors.

Due to the fact that the K. odoratissima species is native to Iran and has many uses, it is important to know the environmental factors affecting the distribution of this species. Also, K. odoratissima species is at risk of extinction due to excessive exploitation, so knowing and introducing the factors affecting it can play an important role in its cultivation and domestication. Also, until now, study that determine the important environmental factors in the distribution of K. doratissima in Iran using BMLR modeling has not been conducted.

-All references should be APA style based.

 Our reply# Our reply# thanks for your attention; done; References have prepared based on the journal format.

Chapter numbering does not follow the journal style.

Our reply# thanks for your attention; done.

  1. Introduction
  2. Results
  3. Discussion
  4. Materials and Methods
  5. Conclusions

Best regards,

Authors,